# Frequency-Aware GAN for Imperceptible Transfer Attack on 3D Point Clouds

## ABSTRACT

With the development of depth sensors and 3D vision, the vulnerability of 3D point cloud models has garnered heightened concern. Almost all existing 3D attackers are deployed in the white-box setting, where they access the model details and directly optimize coordinate-wise noises to perturb 3D objects. However, realistic 3D applications would not share any model information (model parameters, gradients, etc.) with users. Although a few recent works try to explore the black-box attack, they still achieve limited attack success rates (ASR) and fail to generate high-quality adversarial samples. In this paper, we focus on designing a transfer-based black-box attack method, called Transferable Frequency-aware 3D GAN, to delve into achieving a high black-box ASR by improving the adversarial *transferability* while making the adversarial samples more *imperceptible*. Considering that the 3D imperceptibility depends on whether the shape of the object is distorted, we utilize the spectral tool with the GAN design to explicitly perceive and preserve the 3D geometric structures. Specifically, we design the Graph Fourier Transform (GFT) encoding layer in the GAN generator to extract the geometries as guidance, and develop a corresponding Inverse-GFT decoding layer to decode latent features with this guidance to reconstruct high-quality adversarial samples. To further improve the transferability, we develop a dual learning scheme of discriminator from both frequency and feature perspectives to constrain the generator via adversarial learning. Finally, imperceptible and transferable perturbations are rapidly generated by our proposed attack. Experimental results demonstrate that our attack method achieves the highest transfer ASR while exhibiting stronger imperceptibility.

## CCS CONCEPTS

• **Security and privacy**; • **Computing methodologies → Computer vision**;

## KEYWORDS

Point cloud attack, Imperceptibility, Transferability, GAN, Spectral

**ACM Reference Format:**
Anonymous Author(s). 2018. Frequency-Aware GAN for Imperceptible Transfer Attack on 3D Point Clouds. In *Proceedings of Make sure to enter the correct conference title from your rights confirmation emai (Conference acronym 'XX).* ACM, New York, NY, USA, 10 pages. https://doi.org/XXXXXXX.XXXXXXX

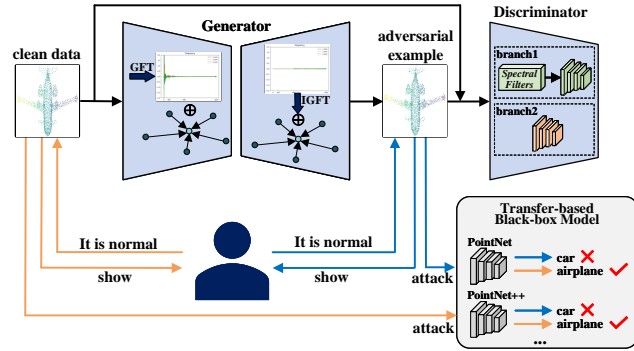

**Figure 1: Illustation of our motivation. We propose to design a generative model to generate high-quality 3D adversarial objects by perceiving the 3D geometric characteristics with spectral information. We also design an adversarial learning scheme to improve the transferability of adversarial samples.**

## 1 INTRODUCTION

Deep Neural Networks (DNNs) are known to be vulnerable to adversarial examples [9, 40], which are indistinguishable from legitimate ones by adding trivial perturbations but often lead to incorrect model prediction. Many efforts have been made into attacks on the 2D image field [6, 22, 30, 44], which often add imperceptible pixel-wise noise on images to deceive the DNNs. However, in addition to image-based 2D attacks, 3D point cloud attacks are also crucial in various safety-critical applications such as autonomous driving [4, 14, 55], robotic grasping [46, 62], medical data analysis [38], and face challenges in realistic scenarios.

Mainstream 3D point cloud attack methods can be roughly divided into two categories: gradient search-based and gradient optimization-based approaches. The gradient search-based works [50, 52, 53, 61] identify critical points from point clouds and modify (add or delete) them to distort the most representative features for misclassification. While the gradient optimization-based works [2, 11, 26, 29, 43, 49, 58] follow the C&W framework [9] to learn to perturb xyz coordinates of each point and generate adversarial samples in an end-to-end manner. Although the above works have achieved high attack success rates, most of them are simply deployed in the white-box setting where the attackers have the full knowledge of victim models including network structure and weights. This setting makes the attacks less practical since most real-world 3D applications will not share their model details with users. Recently, a few works [11, 18, 63] propose to tackle 3D attacks in the black-box setting without using prior model knowledge. However, these works simply utilize geometric distance losses or additional shape knowledge to implicitly constrain the perturbations, failing to properly keep the original 3D object shape and resulting in irregular surfaces or outliers. Therefore, their generated adversarial point clouds are often easily perceivable by humans. Moreover,

these attacks tend to overfit the target network and hardly remain malicious once they are transferred to attack a different victim model.

To this end, in this paper, we focus on addressing the challenging black-box 3D attack from two aspects: improving the adversarial robustness and the imperceptibility of the attack method. Firstly, to improve the adversarial robustness, we aim to force the adversarial point clouds to successfully attack different unknown 3D models. This inspires us to develop a transfer-based attack method, which learns to generate the most harmful noise and make them resist against unknown distortions in unseen models. In this manner, the attack method can achieve high black-box attack success rates. Secondly, to improve the imperceptibility, instead of relying on implicit geometric constraints, we follow the existing generative models, *e.g.*, generative adversarial network (GAN), to explicitly perceive the 3D shape-aware latent characteristics and construct natural/stealthy 3D objects. By carefully guiding the generative model to encode and decode the 3D geometries, it can well generate realistic 3D object shapes. With further adversarial learning design, the generated point cloud can be adversarial to the targets.

Based on the above considerations, we propose a novel Transferable Frequency-aware 3D GAN Attack method, which utilizes spectral tools to explicitly perceive the 3D geometries and encode them in the GAN structure for preserving the 3D structure during the object construction. Specifically, as shown in Figure 1, our GAN-based architecture consists of two components: a frequency-aware generator and a frequency-aware discriminator. We first convert the encoded point-level coordinate/spatial features into the spectral frequencies via graph fourier transform (GFT), then fuse them for feature processing in the frequency-aware generator. This GFT will lead to a compact representation of the geometric data in the spectral domain [15, 16, 34], which is effective in comprehending the shape information of the whole point cloud. During latent feature encoding and decoding in the generator, we perform skip connections of high-frequency features from GFT-based encoder to the IGFT-based decoder of the generator to preserve the detailed geometric contexts of point cloud for making it imperceptible. By using the discriminator to prompt the generator to produce realistic adversarial point clouds, we can improve the imperceptibility of the generated adversarial samples. To further improve the transferability, we extend the traditional discriminator into a dual-branch scheme, which not only utilizes the spatial features to distinguish the point clouds, but also utilizes low-frequency and high-frequency spectral filters to provide additional spectral feature types for adversarial learning. This can train the perturbations to resist possible distortions when transferred to attack unknown models. An additional frozen feature discriminator is also introduced to extract intermediate layer features to maximize feature similarity. In this manner, our proposed attack is able to generate imperceptible 3D adversarial objects with high transferability.

Our main contributions are summarized as follows:

- We make the attempt to improve both the transferability and imperceptibility of black-box attacks on 3D point clouds. To perceive and preserve the geometric characteristics of 3D objects, we develop a frequency-aware GAN model to construct stealthy adversarial point clouds.

- In the GAN-based architecture, we carefully construct a frequency-aware generator with GFT and IGFT designs to perceive and preserve the 3D geometrics for improving the imperceptibility. We also design a dual scheme of discriminator in both spectral and feature contexts to adversarially train the generator for improving the transferability.

- Extensive experiments demonstrate that our attack method achieves the highest transfer attack success rate while exhibiting stronger imperceptibility.

## 2 RELATED WORK

**3D point cloud attacks.** Following previous studies on the 2D image field, many works [43, 50, 52, 53, 60, 61, 63] adapt adversarial attacks into the 3D vision community [3, 10, 31, 32, 45, 47, 51, 54, 56, 59]. [52] proposes point generation attacks by adding a limited number of synthesized points/clusters/objects to a point cloud. [53] utilizes gradient-guided attack methods to explore point modification, addition, and deletion attacks. Their goal is to add or delete key points, which can be identified by calculating the label-dependent importance score referring to the calculated gradient. Recently, more works [26, 43, 58] adopt point-wise perturbation by changing their xyz coordinates, which are more effective and efficient. [26] modifies the FGSM strategy to iteratively search the desired pixel-wise perturbation. [43] adapts the C&W strategy to generate adversarial examples on point clouds and proposes a perturbation-constrained regularization. Besides, [18, 42] propose the query-based black-box attack approach to generate imperceptible perturbations. Several works [7, 20, 23, 35] attack point clouds in the feature space and target at perturbing less points with lighter distortions for an imperceptible adversarial attack. However, the generated adversarial point clouds of all above methods often result in messy distribution or outliers, which are easily perceivable by humans. Meanwhile, these methods tend to overfit the target network, resulting in poor transferability when attacking the victim models.

**Generative models.** Mainstream generative models mainly include generative adversarial networks (GAN) and diffusion models, which have been widely applied in various point cloud fields, including point cloud generation [1, 28, 36, 41], upsampling [27, 33], adversarial attack [11, 63], adversarial defense [39, 57], etc. The GAN models consist of a generator and a discriminator, and the two networks are updated alternatively in an adversarial manner. Unlike GANs, diffusion models gradually introduce noise into the data and then learn how to reverse diffusion process to generate examples. [1] is the first to construct GAN based on a pre-trained auto-encoder to synthesize point clouds indistinguishable from real data. [28] proposes a probabilistic generative model for point clouds by exploiting the reverse diffusion Markov chain to model the distribution of points. [11, 63] train the generator to learn the adversarial distribution and produce adversarial examples by optimizing target loss and GAN loss. [39, 57] utilize diffusion models to restore the perturbed point clouds under various distortions to a clean distribution. Although both models can achieve better performance in terms of generation quality, diffusion models generate datas more slowly than GANs because they require a multi-step iterative process to gradually reduce noise.

Figure 2: Overview of our proposed frequency-aware GAN for imperceptible transfer attack.

# 3 METHOD

## 3.1 Problem Definition

Generally, a point cloud consists of an unordered set of points $P = \{p_i\}_{i=1}^n \in \mathbb{R}^{n \times 3}$ sampled from the surface of a 3D object or scene, where each point $p_i \in \mathbb{R}^3$ is a vector that contains the coordinates $(x, y, z)$ of point $i$, and $n$ is the number of points. In this paper, we mainly focus on attacking the basic point cloud classification task. Given a point cloud $P$ as input, a learned classifier $f(\cdot)$ predicts a vector of confidence scores $f(P) \in \mathbb{R}^C$. The final predicted label is $y = F(P) = argmax_{i \in [C]} f(P)_i \in Y, Y = \{1, 2, 3, ..., C\}$ that represents the class of the source 3D object underlying the point cloud, where $C$ is the number of classes. To attack such classification model, the general objective is to find a perturbation $\Delta \in \mathbb{R}^{n \times 3}$ that generates an adversarial example as $\widetilde{P}$, which misguides the predicted result from the ground truth label $y$ to the other label. We use the $l_p$ norm to regularize the perturbation $\Delta$ to the range $\epsilon$ and formulate the adversarial attack with misclassification loss $L_{mis}$ as the following optimization problem:

$$\max_{\Delta} L_{mis}(F(\widetilde{P}), y), s.t. ||\Delta||_p \leq \epsilon. \tag{1}$$

## 3.2 Overview of Our Attack

**GAN-based architecture.** Previous 3D adversarial attacks generally follow the gradient-based and optimization-based methods, requiring access to the architecture and parameters of the target model, and rely on implicit geometric constraints that limit the

quality of adversarial examples. Besides, their optimization process is also time-consuming and can only optimize perturbation for one specific sample each time. Fortunately, the type of GAN-based method can generate adversarial point clouds that approximate the source point clouds distribution and geometries through adversarial learning, and can quickly generate perturbation for any point cloud after only one training. However, existing GAN attack LG-GAN [63] not only has problems with the loss of details and deformation of adversarial point clouds, resulting in perturbation visible to the human eyes, but also fails to consider the characteristic of attack transferability. Therefore, we propose a new frequency-aware GAN architecture, to generate more difficult imperceptible and transferable adversarial samples. As shown in Figure 2, our proposed GAN-based architecture has two components: a frequency-aware generator $G$ and a dual discriminator $D$. The core is the adversarial training between $G$ and $D$. $G$ tries to generate realistic adversarial point clouds to fool $D$, while $D$ tries to distinguish the generated adversarial point clouds from the source point clouds.

**Our pipeline.** In this paper, we propose a frequency-aware GAN attack method to generate adversarial point clouds with high imperceptibility and transferability. As shown in Figure 2, to improve the imperceptibility, we develop a frequency-aware generator consisting of a GFT-based encoder and a IGFT-based decoder. The encoder extracts high-level feature representation of point clouds, while the decoder maps the feature representation to generate new point clouds. We perform spectral tools of Graph Fourier Transform (GFT) on the encoded features to obtain frequency features for

perceiving the 3D geometric characteristic, which can be divided into low-frequency (LF) and high-frequency (HF) components. We then use these frequency features to preserve the geometric topology by utilizing high-frequency components skip-connections to the decoder to provide more fine-grained geometric guidance for perturbation. To further improve the transferability, we propose a dual discriminator consisting of a frequency-aware discriminator and a feature discriminator. The frequency-aware discriminator is trained to distinguish samples via the low-frequency features and high-frequency features, so that adversarial samples can be trained to be no longer over-fitting to models that solely rely on single spatial features. The feature discriminator directly uses an encoding model to extract features of the intermediate layer for sample distinguishing. By forcing the generator against both two discriminators from different aspects, the generated adversarial samples can achieve high robustness and attack success rate to unknown 3D models.

## 3.3 Improving Imperceptibility with Frequency-aware Generator

The role of the generator is to learn to produce an adversarial point cloud that is as similar as possible to the source point cloud distribution. On this basis, we hope that the generator can further improve the imperceptibility of adversarial point clouds, allowing them to have more fine-grained details of class-related geometric characteristics and not be deformed. To address this issue, we propose a new frequency-aware generator $G$, breaking the limitation of previous generators only being set in the data domain. In particular, during the generator learning, we additionally extract the spectral frequencies of the source point cloud to explicitly explore and capture its geometric structure. Specifically, the spectral domain explicitly reveals the geometric structures from basic shapes to fine details via different frequency bands [13]: the low-frequency components in the frequency domain represent basic shapes, while the high-frequency components encode find-grained details. Therefore, by combining both spatial and frequency features in $G$ with skip connections of the high-frequency components, we are able to guide the generator to comprehend both the semantics and geometrics of the original 3D object for generating imperceptible adversarial samples. The architecture of the frequency-aware generator is shown in the upper of Figure 2.

**Encoder with GFT-based design.** We first introduce the designed GFT-based encoder that encodes both spatial and spectral representations for perceiving the object geometrics. Specifically, to extract high-level representations of point clouds, this encoder utilizes four feature extraction layers (FE), each of which consists of an EdgeConv layer of DGCNN [47] and a graph fourier transform (GFT) [17] layer. We employ the EdgeConv layer to extract the spatial features of point cloud $P = \{p_i\}_{i=1}^n$. In the i-th layer, Edge-Conv uses local structure information to calculate the edge feature set $S$ of each point, and aggregates the features in each set into a new representation of the point. The final spatial representation $P_{spa}^i \in \mathbb{R}^{n \times m}$ of points is obtained by:

$$P_{spa}^i = max_{j:(l,j) \in S}\{ReLU(\boldsymbol{\theta}_m \cdot (\boldsymbol{p}_l - \boldsymbol{p}_j) + \boldsymbol{\varphi}_m \cdot \boldsymbol{p}_j)\},$$
$$j = \{1, 2, ..., n\},$$
(2)

where $m$ is the feature dimension of the output, $\boldsymbol{\theta}_m$ and $\boldsymbol{\varphi}_m$ are the weight of filters.

In addition to extracting the spatial features, we also design spectral neural modules to encode frequency features for perceiving the geometric characteristics. Given a point cloud $P$, we construct a K-nearest-neighbor (KNN) graph for $P$, and calculate Laplacian matrix [37] as $L$. $L$ admits an eigen-decomposition $L = U\Lambda U^T$, where $U = [\boldsymbol{u}_1, ..., \boldsymbol{u}_n]$ is an orthonormal matrix containing the eigenvectors $\boldsymbol{u}_i$ which is called the graph fourier basis, and $\Lambda = diag(\lambda_1, ..., \lambda_n)$ consists of eigenvalues (i.e. frequencies) $\{\lambda_1 = 0 \leq \lambda_2 \leq ... \leq \lambda_n\}$. The coordinates of points in $P$ are treated as graph signals, and its GFT coefficient vector $\hat{P}$ is defined as [12]:

$$\hat{P} = \phi_{GFT}(P) = U^T P.$$
(3)

In the i-th layer, we utilize the GFT layer to convert the output $P_{spa}^i$ of the EdgeConv layer into frequency features as $\hat{P}_{fre}^i = \phi_{GFT}(P_{spa}^i)$. We then perform tenor addition on the spatial features and frequency features to preserve the geometric topology and semantics as $P_{spa}^{i+1} = P_{spa}^i + \hat{P}_{fre}^i$. Finally, the features of each FE layer are concatenated to obtain the high-level global features as:

$$features = conv([P_{spa}^1, P_{spa}^2, P_{spa}^3, P_{spa}^4]).$$
(4)

**Decoder with IGFT-based design.** To generate adversarial point clouds with fine-grained details of class-related geometric characteristics and less noise, we put the high-frequency components of the GFT-based encoder into the decoder via skip connections, thus making the adversarial samples less distinguishable by human eyes. This IGFT-based decoder also mainly contains four feature reconstruction layers (FR), each of which consists of the conv layer and the IGFT layer. Specifically, in the i-th FR layer, the global features generated by the encoder are reconstructed using conv layer to obtain features as $A = conv_i(features)$. The IGFT layer performs the high-frequency components from the $4 - i + 1$ layer of the encoder to obtain features as $B = \phi_{IGFT}((\hat{P}_{fre}^{4-i+1})_{HF})$. The IGFT is defined as:

$$P = \phi_{IGFT}(\hat{P}) = U\hat{P}.$$
(5)

These two features are combined as $A + B$ and sent to the FR layer. Finally, the fully connected layer is utilized to generate perturbation for the point cloud.

## 3.4 Improving Transferability with Dual Discriminator

To improve the transferability of adversarial samples, we aim to build a powerful discriminator. Through alternate training of the generator and the discriminator, GAN finally reaches a balance, which is that the generator generates sufficiently realistic adversarial point clouds, while the discriminator has difficulty distinguishing the adversarial point clouds from the source point clouds. The insight is that, if the generated adversarial point clouds can fool a powerful discriminator model, they can also fool other unknown 3D classification models. Therefore, in this section, we propose a dual discriminator consisting of a frequency-aware discriminator $D_\psi$ and a feature discriminator $D_\gamma$, to push the whole GAN generating transferable adversarial samples. This discriminator will assist in improvement from two aspects: (1) enriching the feature types in

spectral frequency representations instead of only relying on spatial features; (2) learning to distinguish the feature similarity from the intermediate layer. The architecture of the dual discriminator is shown in the bottom of Figure 2.

**Frequency-aware discriminator.** We propose a novel frequency-aware discriminator that can learn not only spatial features, but also low-frequency and high-frequency spectrum domain representations to better distinguish point clouds, and enrich the feature types to improve transferability. This discriminator consists of three modules with a similar architecture but different inputs. The modules contain multi-layer perceptrons (MLPs) and two GFT layers. The MLPs module $f_{pc}(\cdot)$ takes point cloud as input to extract spatial features. The low-pass GFT module $f_{lf}(\cdot)$ first utilizes a low-pass filter to obtain low-frequency (LF) points, which are then used as input to extract the low spectral frequency representations. Similarly, the high-pass GFT module $f_{hf}(\cdot)$ utilizes a high-pass filter to obtain high-frequency (HF) points, and extracts high spectral frequency features.

We first define the graph filter as follows to allow or suppress the passage of frequency components:

$$P' = H(\lambda)P = U \begin{bmatrix} h(\lambda_1) & & \\ & \ddots & \\ & & h(\lambda_n) \end{bmatrix} U^T P, \qquad (6)$$

where $h(\lambda_i(i = 1, 2, ..., n))$ denote the frequency response of a graph filter. The low-pass filter $h(\lambda_i)$ is as follows:

$$h(\lambda_i) = \begin{cases} 1, & i < b, \\ 0, & i \geq b. \end{cases} \qquad (7)$$

In contrast, the high-pass filter $h(\lambda_i)$ is as follows:

$$h(\lambda_i) = \begin{cases} 0, & i < b, \\ 1, & i \geq b. \end{cases} \qquad (8)$$

Here, $b$ is the dividing point between low-frequency band and high-frequency band. Referring to the previous work [25], we set $b = 100$ in the experiments.

Specifically, the source point cloud $P$ and adversarial point cloud $\widetilde{P}$ are respectively input into the low-pass filter and high-pass filter to get LF points $(P')_L, (\widetilde{P}')_L$ and HF points $(P')_H, (\widetilde{P}')_H$. Then the point clouds, LF points and HF points are input into the corresponding modules to get their respective spatial and spectral features. Finally, all types of features are concatenated to form the input fed into the final discriminant vector, and the corresponding confidence values are generated through three FC layers as:

$$D_\psi(P) = FC([f_{pc}(P), f_{lf}((P')_L), f_{hf}((P')_H)], \qquad (9)$$

$$D_\psi(\widetilde{P}) = FC([f_{pc}(\widetilde{P}), f_{lf}((\widetilde{P}')_L), f_{hf}((\widetilde{P}')_H)]). \qquad (10)$$

**Feature discriminator.** Further, we construct a feature discriminator to make a distinction between source point clouds and adversarial point clouds from the perspective of feature similarity. The intermediate features extracted by different models during classification show strong similarities. This inspires us to utilize a frozen discriminator as the feature extractor to guide the frequency-aware generator to produce transferable adversarial point clouds via a feature distance loss. Specifically, the feature discriminator is a pre-trained classifier. We respectively input the point cloud

$P$ and $\widetilde{P}$ into the $D_\gamma$, extract the corresponding features $r_{ori}$ and $r_{adv}$ from intermediate layer outputs of $D_\gamma$, calculate the distance between $r_{ori}$ and $r_{adv}$, and optimize the generator by maximizing the distance to destroy feature similarity.

## 3.5 Objective Loss Function

**Adversarial Loss.** The goal of the $D_\psi$ is to correctly differentiate between the source point cloud $P$ and the adversarial point cloud $\widetilde{P}$, while the goal of the $G$ is to fool $D_\psi$. That is, adversarial loss $L_{dis}$ of $D_\psi$ is to maximize the probability that the $D_\psi$ identifies $P$ as true and $\widetilde{P}$ as false. On the contrary, adversarial loss $L_{gen}$ of the $G$ is to maximize the probability that the $D_\psi$ identifies $\widetilde{P}$ as true. Based on this and previous work [41], we adopt the following adversarial loss:

$$L_{dis}(P, \widetilde{P}) = -\mathbb{E}_{P \sim \mathbb{P}_P}[D_\psi(P)] + \mathbb{E}_{\widetilde{P} \sim \mathbb{P}_{\widetilde{P}}}[D_\psi(\widetilde{P})], \qquad (11)$$

$$L_{gen}(\widetilde{P}) = -\mathbb{E}_{\widetilde{P} \sim \mathbb{P}_{\widetilde{P}}}[D_\psi(\widetilde{P})], \qquad (12)$$

where $\mathbb{P}_P$ is the distribution of $P$ and $\mathbb{P}_{\widetilde{P}}$ is the distribution of $\widetilde{P}$.

**Reconstruction Loss.** As with the adversarial loss objective, to make $\widetilde{P}$ similar to $P$, we adopt the L2-norm distance as reconstruction loss $L_{rec}$ [63]:

$$L_{rec}(P, \widetilde{P}) = \sqrt{\sum_{p_i \in P, \widetilde{p}_i \in \widetilde{P}} (p_i - \widetilde{p}_i)^2}. \qquad (13)$$

**Feature Loss.** We introduce feature loss $L_{fea}$ by maximizing the distance between features $r_{ori}$ and $r_{adv}$ from intermediate layer outputs of $D_\gamma$ to optimize the $G$. This makes the adversarial features far away from the clean features, and the final classification is biased towards another category. Mathematically, the $L_{fea}$ is defined as follows:

$$L_{fea}(r_{ori}, r_{adv}) = -||r_{ori} - r_{adv}||_2^2. \qquad (14)$$

In summary, both generator $G$ and discriminator $D_\psi$ are optimized during training using the above losses as:

$$L_G = L_{gen} + w_{rec}L_{rec} + w_{fea}L_{fea}, \qquad (15)$$

$$L_D = L_{dis}, \qquad (16)$$

where $w_{rec}$ and $w_{fea}$ are the weight factors.

## 3.6 Iterative Learning Generator and Discriminator

In our attack method, the frequency-aware generator and the dual-discriminator utilize the adversarial learning process to improve their respective performance, prompting the generation of adversarial samples. Specifically, by minimizing the loss functions, we iteratively update the parameters $\eta$ of the frequency-aware generator $G$ and the parameters $\zeta$ of the frequency-aware discriminator $D_\psi$ via gradient descent. Prior to this, we use random mask proposed by [48] to screen out meaningless perturbation points. This mask $M$ is the same size as the perturbation and consists of 0 or 1. We perform matrix multiplication of the mask with the perturbation to randomly discard the perturbation points. The overall training procedure of our model is summarized in Algorithm 1. During the inference, we directly utilize the generator $G$ to quickly create imperceptible and transferable adversarial point cloud.

**Table 1: Comparative results on the perturbation sizes of adversarial point clouds generated by different attack methods across different 3D classification models under** 100% **ASR.**

| Setting | Attack Methods | PointNet | | | PointNet++ | | | DGCNN | | | PCT | | |
|---------|---------------|----------|----------|----------|----------|----------|----------|----------|----------|----------|----------|----------|----------|
| | | $D_h$ | $D_c$ | $D_{norm}$ | $D_h$ | $D_c$ | $D_{norm}$ | $D_h$ | $D_c$ | $D_{norm}$ | $D_h$ | $D_c$ | $D_{norm}$ |
| White-Box | FGSM [53] | 0.1853 | 0.1326 | 0.7936 | 0.2275 | 0.1682 | 0.8357 | 0.2506 | 0.1890 | 0.8549 | 0.2478 | 0.1815 | 0.8493 |
| | PGD [30] | 0.1322 | 0.1329 | 0.7384 | 0.1623 | 0.1138 | 0.7596 | 0.1546 | 0.1421 | 0.7756 | 0.1625 | 0.1388 | 0.7541 |
| | AdvPC [11] | 0.0343 | 0.0697 | 0.6509 | 0.0429 | 0.0685 | 0.6750 | 0.0148 | 0.0623 | 0.6304 | 0.0369 | 0.0625 | 0.6806 |
| | 3D-ADV [52] | 0.0105 | 0.0003 | 0.5506 | 0.0381 | 0.0005 | 0.5699 | 0.0475 | 0.0005 | 0.5767 | 0.0237 | 0.0007 | 0.6011 |
| | LG-GAN [63] | 0.0362 | 0.0347 | 0.7184 | 0.0407 | 0.0238 | 0.6896 | 0.0348 | 0.0119 | 0.8527 | 0.0312 | 0.0419 | 0.9416 |
| | GeoA [49] | 0.0175 | 0.0064 | 0.6621 | 0.0357 | 0.0198 | 0.6909 | 0.0402 | 0.0176 | 0.7024 | 0.0428 | 0.0217 | 0.7350 |
| Black-Box | SI-Adv [18] | 0.0431 | 0.0003 | 0.9351 | 0.0444 | 0.0003 | 1.0857 | 0.0336 | 0.0004 | 0.9081 | 0.0518 | 0.0004 | 0.9741 |
| Transfer-based Black-Box | Ours | **0.0223** | **0.0003** | **0.8952** | **0.0190** | **0.0006** | **0.9757** | **0.0177** | **0.0005** | **0.9561** | **0.0194** | **0.0005** | **0.8436** |

## 4 EXPERIMENTS

### 4.1 Dataset and 3D Models

**Dataset.** We use ModelNet40 [51] in our experiments to evaluate the attack performance. ModelNet40 consists of 12,311 CAD models from 40 object categories, in which 9,843 models are intended for training and the other 2,468 for testing. Following the settings of previous work [13, 24, 49], we uniformly sample 1024 points from the surface of each object and scale them into a unit ball.

**3D Models.** Following previous works, we select four commonly used point cloud classification networks in 3D computer vision community as the feature discriminator and the victim models, i.e., PointNet [31], PointNet++ [32], DGCNN [47] and PCT [10]. We generate the adversarial point clouds over each of them, and further evaluate the transferability of our proposed attack among them.

---

**Algorithm 1** Training of the proposed frequency-aware GAN

---

**Require:** The source point clouds $P$, pre-trained feature discriminator $D_\gamma$, the number of iteration $T$, and maximum perturbation magnitude $\epsilon$.

**Ensure:** Frequency-aware generator $G$

1: Initialize the frequency-aware generator $G$ and the frequency-aware discriminator $D_\psi$.
2: **for** $i = 1, \ldots, T$ **do**
3:      Get batches of source point clouds $P$
4:      Get the adversarial perturbations $\Delta = clip(G(P))$ which is restricted to a range of $[-\epsilon, \epsilon]$
5:      Get the adversarial point clouds with random mask $\widetilde{P} = P + M \times \Delta$
6:      Get the confidence values $D_\psi(P), D_\psi(\widetilde{P})$
7:      Get intermediate feature maps $r_{ori}$ and $r_{adv}$ with $D_\gamma$
8:      Calculate loss of $D_\psi$:
9:         $L_D = -\mathbb{E}_{P \sim \mathbb{P}_P}[D_\psi(P)] + \mathbb{E}_{\widetilde{P} \sim \mathbb{P}_{\widetilde{P}}}[D_\psi(\widetilde{P})]$
10:     Update $\zeta \leftarrow \min L_D$
11:     Calculate reconstruction loss $L_{rec}$ with L2-norm distance
12:     Calculate feature loss $L_{fea} = -||r_{ori} - r_{adv}||_2^2$
13:     Calculate distinction loss $L_{gen} = -\mathbb{E}_{\widetilde{P} \sim \mathbb{P}_{\widetilde{P}}}[D_\psi(\widetilde{P})]$
14:     Calculate loss of $G$:
15:        $L_G = L_{rec} + L_{fea} + L_{gen}$
16:     Update $\eta \leftarrow \min L_G$
17: **return** $G$

---

### 4.2 Experimental Settings

**Evaluation Metrics.** To quantitatively evaluate the effectiveness of our proposed attack, we measure the generated adversarial examples by the attack success rate (ASR), L2-norm distance $D_{norm}$ [5], Chamfer distance $D_c$ [8] and Hausdorff distance $D_h$ [19].

**Implementation Details.** We train the model for 100 epochs using the Adam [21] optimizer with a batch size of 32, and the learning rate of $G$ and $D_\psi$ are both 0.0001. We set K = 10 to build a KNN graph and set the perturbation threshold as $\epsilon = 0.16$. The weight of reconstruction loss $w_{rec}$ is set to 1. The weight of the feature loss $w_{fea}$ and the intermediate layer are different due to the different models used by the feature discriminator. In PointNet, PointNet++, DGCNN, and PCT, $w_{fea}$ is set to 9, 10, 4, 15 respectively, and intermediate layer is set to 6th, 5th, 8th, 9th. All experiments are implemented on a single NVIDIA RTX 5000 GPU.

### 4.3 Evaluation on the Imperceptibility

**Comparison with existing methods.** To investigate the imperceptibility of our attack, we measure the perturbation sizes of different adversarial point clouds required to achieve 100% of attack success rate with three evaluation metrics. As shown in Table 1, our attack achieves smaller perturbation sizes than the black-box model and achieves very competitive results with white-box models. Compared with methods such as 3D-ADV [52] that modify a few points, our method has a slightly higher $D_{norm}$ because we use GAN to conduct global perturbations. Perturbing each point can alter the distance to the original point, thereby increasing the $D_{norm}$ value for precise point-to-point distance measurement as the number of perturbed points increases. However, our attack performs better in $D_c$ and $D_h$ since we utilize the spectral tool to capture and preserve the geometric structures of the point cloud.

**Visualization results.** We provide visualization on adversarial samples generated by our attack, LG-GAN [63] and SI-Adv [18] in Figure 3. Compared to LG-GAN, our adversarial samples have less shape distortion. Besides, our method also mitigates detail loss and the outlier points, thereby generating imperceptible adversarial samples. More visualizations are in the supplementary.

### 4.4 Evaluation on the Transferability

To investigate the transferability of our attack, we first utilize the feature discriminator as the white-box model. Then we craft adversarial samples on different feature discriminator and test them

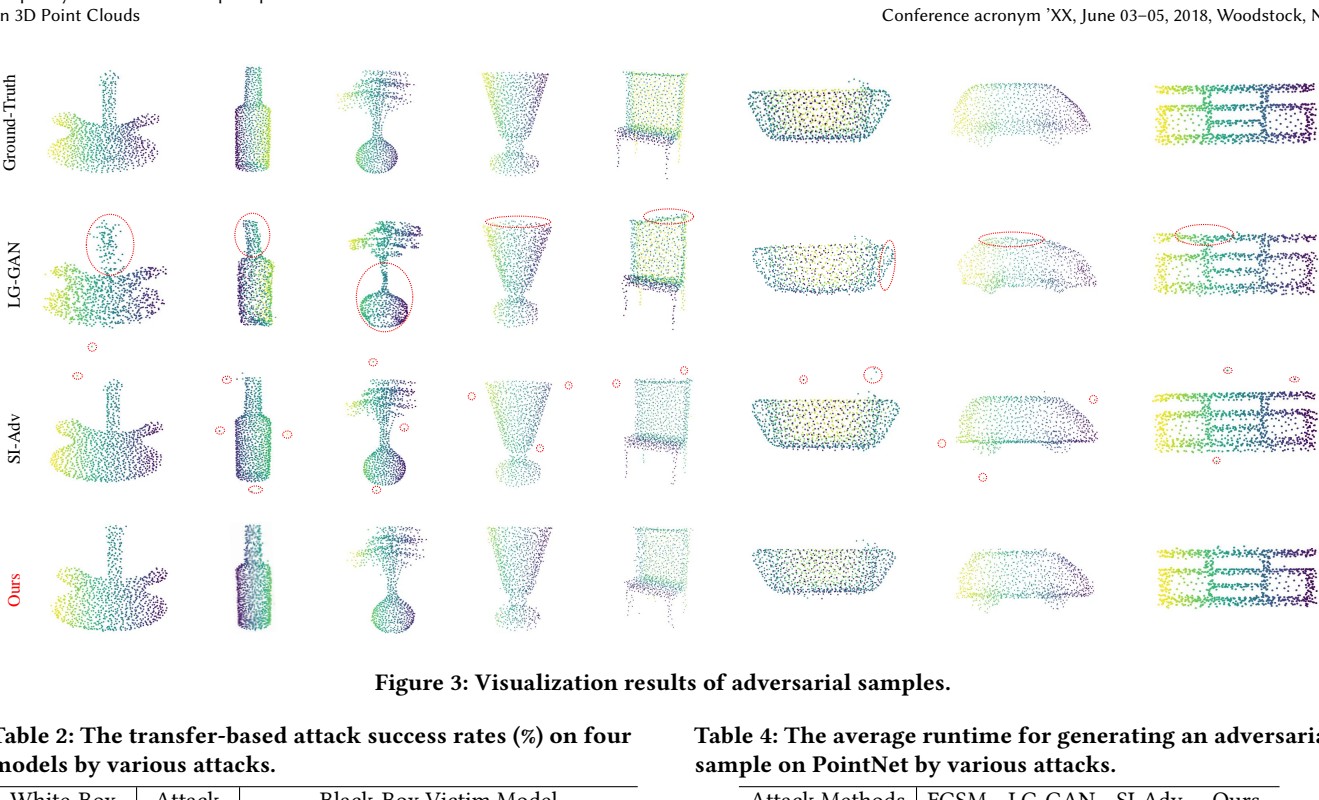

**Figure 3: Visualization results of adversarial samples.**

**Table 2: The transfer-based attack success rates (%) on four models by various attacks.**

| White-Box Target Model | Attack Methods | Black-Box Victim Model | | | |
|---|---|---|---|---|---|
| | | PointNet | PointNet++ | DGCNN | PCT |
| PointNet | FGSM | **100.0** | 3.9 | 0.6 | 3.7 |
| | LG-GAN | 98.3 | 11.6 | 14.5 | 12.9 |
| | SI-Adv | **100.0** | 7.1 | 14.1 | 8.5 |
| | Ours | **100.0** | **84.4** | **68.4** | **71.2** |
| PointNet++ | FGSM | 3.2 | **100.0** | 5.6 | 10.0 |
| | LG-GAN | 10.2 | 93.5 | 9.1 | 18.4 |
| | SI-Adv | 21.4 | 94.4 | 10.5 | 17.1 |
| | Ours | **62.4** | **100.0** | **70.8** | **73.2** |
| DGCNN | FGSM | 3.6 | 7.2 | **100.0** | 8.9 |
| | LG-GAN | 5.9 | 17.5 | 97.3 | 14.2 |
| | SI-Adv | 15.8 | 11.4 | **100.0** | 25.4 |
| | Ours | **55.2** | **73.6** | **100.0** | **78.4** |
| PCT | FGSM | 9.1 | 7.6 | 12.3 | **100.0** |
| | LG-GAN | 17.6 | 24.3 | 27.8 | **100.0** |
| | SI-Adv | 27.9 | 18.4 | 31.7 | 95.1 |
| | Ours | **72.4** | **81.6** | **74.4** | **100.0** |

**Table 3: The attack success rates (%) on PointNet++ by various attacks under defense.**

| Defense \ Attack | None | FGSM | LG-GAN | SI-Adv | Ours |
|---|---|---|---|---|---|
| No Defense | 0.0 | **100.0** | **100.0** | **100.0** | **100.0** |
| SRS | 0.9 | 9.7 | 38.5 | 70.1 | **73.2** |
| SOR | 0.6 | 6.3 | 64.2 | 78.9 | **79.6** |

on the four models we consider. The corresponding transfer-based attack success rates are shown in Table 2. We can observe that the ASRs of all attack methods are very close to 1 when the black-box victim model is the same as the white-box target model. When the victim model is different from the target model, we can find that

**Table 4: The average runtime for generating an adversarial sample on PointNet by various attacks.**

| Attack Methods | FGSM | LG-GAN | SI-Adv | Ours |
|---|---|---|---|---|
| Runtime | 0.082s | 0.04s | 0.58s | **0.006s** |

our attack achieves higher ASRs than other methods. The main reason is that, the proposed dual-branch discriminator from both frequency and feature perspectives through adversarial learning encourages the genarator to produce the most harmful perturbations, which can resist possible distortions when transferred to attack unknown models.

### 4.5 Evaluation on Existing Point Cloud Defense

To evaluate the robustness of our attack against different adversarial defenses, we conduct the experiments on two widely used defense methods: Simple Random Sampling (SRS) [53] and Statistical Outlier Removal (SOR) [64]. As shown in Table 3, (1) the ASRs on clean data is equal or close to 0 perhaps due to the modification of point clouds by the defense methods. (2) The FGSM attack has lower ASRs since it shifts points along xyz directions without any constraint of the length. This easily results in uneven local distribution and outliers. (3) Our method achieves better ASRs than other attacks under the defenses. This is because the adversarial sample we generate is highly similar to the original one in both geometric topology and point distributions. Meanwhile, our attack alleviates the outlier point problems.

### 4.6 Evaluation on the Runtime

We assess the average runtime required to generate an adversarial point cloud in the inference phase across various attack methods. As shown in Table 4, our method demonstrates a quicker execution compared to others. The FGSM attack accesses the target model for one forward-propagation and one back-propagation processes. The generator architecture of LG-GAN based on PointNet++ is more

**Table 5: Comparison with baseline diffusion model on transfer-based attack success rates (%) and runtime (s). Target model: PointNet.**

| Methods | PointNet | PointNet++ | DGCNN | PCT | Runtime |
|---------|----------|------------|-------|-----|---------|
| Diffusion | 76.2 | 19.1 | 34.8 | 35.7 | 0.81 |
| Ours | **100** | **84.4** | **68.4** | **71.2** | **0.006** |

**Table 6: Effects of GFT and IGFT ablation in the frequency-aware generator on imperceptibility on PointNet.**

| GFT | IGFT | | $D_h$ | $D_c$ | $D_{norm}$ |
|-----|------|--|-------|-------|------------|
|  | high-frequency | low-frequency | | | |
| ✓ | ✓ | ✗ | **0.0223** | **0.0003** | **0.8952** |
| ✓ | ✗ | ✓ | 0.0272 | 0.0009 | 1.4143 |
| ✓ | ✓ | ✓ | 0.0263 | 0.0006 | 1.1975 |
| ✓ | ✗ | ✗ | 0.0276 | 0.0005 | 1.1085 |
| ✗ | ✗ | ✗ | 0.0506 | 0.0019 | 1.9649 |

**Table 7: Sensitivity on hyper-parameter $b$ on PointNet.**

| Metric \ Variant | $b = 50$ | $b = 100$ | $b = 200$ | $b = 400$ |
|------------------|----------|-----------|-----------|-----------|
| $D_h$ | 0.0243 | **0.0223** | 0.0228 | 0.0238 |
| $D_c$ | 0.0004 | **0.0003** | 0.0004 | 0.0004 |
| $D_{norm}$ | 0.9607 | **0.8952** | 0.9011 | 0.8988 |

complex than ours. The SI-Adv attack requires multiple queries to the target model. While our attack only performs one forward-propagation process on generator to create an adversarial sample.

## 4.7 Comparison with Diffusion Model

GANs and diffusion models are both generative models designed to learn from training data and generate new data samples. Currently, there is no research on point cloud adversarial attack based on diffusion model. Therefore, we conduct a comparative analysis with the baseline diffusion model. As show in Table 5, our attack employs the GAN-based approach, which not only achieves superior ASRs but also accelerates the generation of adversarial examples.

## 4.8 Ablation Study

**The influence of frequency features and skip connections in frequency-aware generator.** To verify the effect of these factors in making the adversarial samples imperceptible, we evaluate five ablations in frequency-aware generator: (1) our method; (2) replacing high-frequency (HF) skip connections with low-frequency (LF) skip connections; (3) using HF and LF skip connections; (4) removing HF skip connections; (5) removing the GFT and HF skip connections. As show in Table 6, our method achieves the smallest perturbations than other strategies in all metrics. This is because the frequency features perceive the geometric topology and their high-frequency components can encode find-grained details well.

**The influence of dual discriminator.** To verify the impact that the feature discriminator and the frequency-aware discriminator work together to improve the transferability, we design four ablations in dual discriminator: (1) only the feature discriminator $D_\gamma$; (2) only the the frequency-aware discrimintor $D_\psi$; (3) neither $D_\gamma$ nor $D_\psi$; (4) both $D_\gamma$ and $D_\psi$. As show in Figure 4, the dual-branch strategy realizes the highest ASRs since we construct a powerful

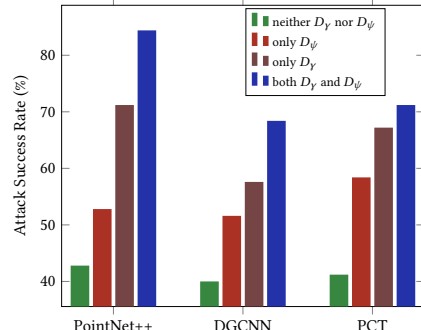

**Figure 4: Effects of each component in the dual discriminator on transferability. The white-box target model is PointNet and the x-axis is the black-box victim models.**

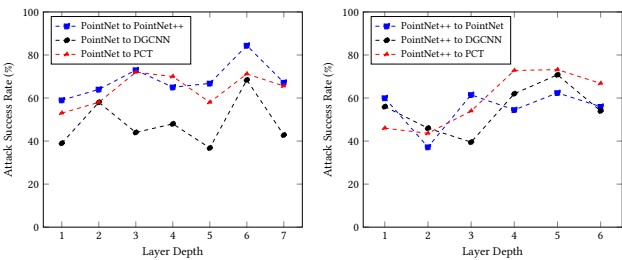

Feature Discriminator: PointNet     Feature Discriminator: PointNet++
**Figure 5: Effects of feature layer selection on transferability.**

discriminator to adversarially train the generator from frequency and feature perspectives.

**The influence of intermediate layer selection in feature discriminator.** Using features of different depths to optimize the generator produces adversarial samples that have different ASRs when attacking the black-box victim models. Based on the experimental results in Figure 5, we select the optimal intermediate layer for each feature discriminator to extract the features.

**Sensitivity on hyper-parameter $b$.** The hyper-parameter $b$ represents the dividing point between the low-frequency band and the high-frequency band. First, it is used in frequency-aware generator to improve imperceptibility and we verify its impact in Table 7. When $b$ is set to 100, our attack achieves the smallest perturbations. Futhermore, $b$ is also used in frequency discriminator, so we perform ablation to test its impact on transferability. The results are in the supplementary.

## 5 CONCLUSIONS

In this paper, we propose a novel Transferable Frequency-aware 3D GAN method, to attack the 3D models in a challenging transferable black-box setting. Specifically, we utilize the spectral tool in the GAN architecture with GFT and IGFT layer designs to perceive and preserve the geometric structures of the 3D point cloud to improve imperceptibility. Besides, we also develop a dual learning scheme of discriminator from both frequency and feature perspectives to constrain the generator via adversarial learning. In this manner, our proposed attack method is able to generate imperceptible 3D adversarial objects with high transferability. Experiments demonstrate the effectiveness of the proposed attack.

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
