# OpenReview forum: "Frequency-Aware GAN for Imperceptible Transfer Attack on 3D Point Clouds"
_acmmm.org/ACMMM/2024/Conference — MM2024 Poster_

### Official Review · Reviewer_K6HU · 2024-05-19

**Rating:** 4
**Confidence:** 3

**Summary:**

The paper presents a novel method called Frequency-Aware GAN for imperceptible transfer attacks on 3D point clouds. This approach leverages the Graph Fourier Transform (GFT) to manipulate point cloud data imperceptibly and ensure the perturbations are transferable across different models. By integrating both frequency and spatial features, the method aims to generate adversarial point clouds that are both effective in attacks and less detectable by human observers or automated systems.

**Strengths:**

1. Utilizing frequency components in the GAN framework helps maintain the geometric integrity of the point clouds, making the adversarial examples hard to detect visually.
2.The dual learning approach with a frequency-aware discriminator enhances the ability of adversarial examples to perform across various unknown 3D models, making it robust in black-box scenarios.

**Limitations:**

The method proposed in this paper is intriguing; however, I have the following concerns:

Firstly, the computational cost introduced by the proposed method should be considered in the experiments, and further comparisons are needed.

Secondly, compared to other methods, the proposed approach does not seem to make significant advancements. Are there additional experiments that could demonstrate the effectiveness of the method?

Lastly, it would be beneficial to include more visual results that intuitively demonstrate the advantages of the method. If these cannot be provided in the response, I hope the authors can incorporate them into the article. Currently, it is challenging to understand the effective work done from the article, and I suggest that the authors enhance the readability of the paper.

If my concerns are addressed, I would consider raising the score.

**Suitability:**

3

---

### Official Review · Reviewer_ghmb · 2024-05-24

**Rating:** 5
**Confidence:** 3

**Summary:**

This article introduces a novel method for enhancing the security of 3D point cloud models. Specifically, this method aims to explore the inherent vulnerabilities of 3D models under more practical scenarios by significantly improving black-box attack success rates (ASR) while ensuring the adversarial samples remain imperceptible. To achieve this, the authors integrate a Graph Fourier Transform (GFT) encoding layer within the GAN generator to precisely capture and retain the 3D geometric structures of objects. An Inverse-GFT decoding layer complements this by reconstructing high-quality adversarial samples guided by the extracted geometries. Additionally, the study introduces a dual learning scheme for the discriminator, optimizing it from both frequency and feature perspectives to refine the generative adversarial network's output. This results in the creation of imperceptible, yet highly transferable, perturbations.

**Strengths:**

1. This article is well written and clearly organized.
2. The article's ingenious design of the GAN network to balance black-box attack performance with the idea of imperceptibility is interesting and holds greater practical value.
3. The experiments presented in this article are comprehensive, particularly the comparison with diffusion models, which effectively alleviates my concerns about the efficacy of generating results using GAN methods.

**Limitations:**

1. [More Verification on Defense Methods] The current validation methods assessing the efficacy of FAGAN against defense mechanisms seem to be somewhat limited. It might be better to include additional defense methods, such as DUP-Net defense and IF-Defense.
2. [Extend to the Physical World] The experiments described in this article are exclusively conducted within a digital framework. It would be compelling if the authors could extend their validations to include physical world scenarios. Doing so could potentially strengthen the effectiveness of your method and enhance FAGAN's practical applicability.

**Suitability:**

2

---

### Official Review · Reviewer_mCsp · 2024-05-27

**Rating:** 2
**Confidence:** 3

**Summary:**

The paper proposes a novel method for generating adversarial point clouds that are both imperceptible and highly transferable. The GAN model presented in this paper enhances the imperceptibility of the generated adversarial samples through frequency-domain perception and improves the transferability of the adversarial samples through dual discrimination in both frequency domain and feature space. The authors demonstrate the effectiveness of the proposed method through experiments.

**Strengths:**

(1)	By incorporating frequency-domain perception, the generator preserves the geometric features of 3D point clouds, resulting in adversarial samples that are difficult to detect.
(2)	With dual discrimination in both frequency domain and feature space, the generator produces adversarial samples that effectively attack different models.
(3)	The method quickly generates adversarial point clouds with only one forward propagation process, significantly reducing the generation time.

**Limitations:**

(1)	Unconvincing Motivation: The paper suggests that frequency-domain information and generative models can simultaneously enhance both the transferability and imperceptibility of adversarial samples. However, these two goals have long been considered a trade-off, making it difficult to achieve both simultaneously. The motivation in the paper is not convincingly presented, and the writing needs refinement as the logical flow is not smooth.
(2)	High-Frequency Information Skip Connection: The paper mentions using skip connections of high-frequency information to enhance the imperceptibility of adversarial samples, but it lacks detailed theoretical or intuitive explanation. It is unclear why this operation can effectively improve imperceptibility. The human visual system is more sensitive to low-frequency information, and the paper's focus on high-frequency information might overlook this important factor, leading to adversarial samples that are still easily detectable in some cases?
(3)	Too Many Model Components and Insufficient Ablation Studies: The paper's model includes multiple components, but the ablation studies are not sufficient. Specifically, the lack of ablation experiments for the dual discriminator does not show the specific contribution of each component to the overall performance.
(4)	Insufficient Comparison with Typical Point Cloud Transfer Attack Methods: The paper does not sufficiently compare with some typical point cloud transfer attack methods (such as advPC, AOF, SS-attack), making it difficult to comprehensively demonstrate the advantages and disadvantages of the proposed method.

**Suitability:**

2

---

### Meta-Review · Area_Chair_agFe · 2024-06-27

**Recommendation:** Accept (Poster)
**Confidence:** 5

**Metareview:**

This paper incorporates frequency-domain perception to point cloud GANs, which is interesting and effective. All the reviewers agree to accept this paper. I recommend a decision of acceptance. In the final version, the authors should further explain the motivation of the paper as well as including some key experiments in the rebuttal.